# Profiles of Problematic Internet Use and Its Impact on Adolescents’ Health-Related Quality of Life

**DOI:** 10.3390/ijerph16203877

**Published:** 2019-10-13

**Authors:** Juan M. Machimbarrena, Joaquín González-Cabrera, Jéssica Ortega-Barón, Marta Beranuy-Fargues, Aitor Álvarez-Bardón, Blanca Tejero

**Affiliations:** 1Faculty of Psychology, University of the Basque Country (UPV/EHU), 20018 Donostia-San Sebastian, Spain; juanmanuel.machimbarrena@ehu.eus; 2Faculty of Education, Universidad Internacional de la Rioja (UNIR), 26006 Logrono, Spain; jessica.ortega@unir.net (J.O.-B.); marta.beranuy@unir.net (M.B.-F.); aitor.alvarez@unir.net (A.Á.-B.); blanca.tejero@unir.net (B.T.)

**Keywords:** problematic internet use, health-related quality of life, adolescent, cut-off point, internet addiction

## Abstract

The internet has been a breakthrough for adolescents in many ways, but its use can also become dysfunctional and problematic, leading to consequences for personal well-being. The main objective is to analyze profiles related to problematic internet use and its relationship with health-related quality of life (HRQoL). An analytical and cross-sectional study was carried out in a region of northern Spain. The sample comprised 12,285 participants. Sampling was random and representative. Mean age and standard deviation was 14.69 ± 1.73 (11–18 years). The Spanish versions of the Problematic and Generalized Internet Use Scale (GPIUS2) and of the Health-Related Quality of Life (KIDSCREEN-27) were used. Four profiles were detected (non-problematic use, mood regulator, problematic internet use, and severe problematic use). The prevalence of these last two profiles was 18.5% and 4.9%, respectively. Problematic internet use correlated negatively and significantly with HRQoL. The severe problematic use profile presented a significant decrease in all dimensions of HRQoL. Analyses were carried out to extract a cut-off point for GPIUS2 (52 points). The results and practical implications are discussed.

## 1. Introduction

Digital society is a source of development and opportunities for people, because it allows them to communicate, obtain information, and develop projects in a way that was previously unthinkable. However, despite all the advantages it implies, digital society also poses risks, especially for younger people. Currently, adolescents live in a very different way from other generations, and young people have a complex and bidirectional relationship between what is occurring online and offline, and, according to the model of co-construction, adolescents shape their reality by connecting their offline and online worlds, the latter frequently predominating [1]. People born since the year 2000 have received the name of the Z-generation, the postmillennial or centennial generation, and one of the elements that has clearly differentiated them from other generations is the massive use of internet on mobile phones [2]. This can be seen in international surveys that reveal that up to 24% of adolescents between 13 and 17 years of age are constantly connected to the internet [3] or in the Spanish context, where it has been found that, among more than 25,000 adolescents between 11 and 18 years, almost 96% own a smartphone, and of them, 87% claim to use it daily [4]. This higher exposure increases the chances that many of the classic psychosocial problems (abuse, violence, and addictions) will have quickly found their online equivalent (cyberbullying, sexting, problematic internet use, nomophobia, etc.), and their prevalence is increasing [1,5,6].

Currently, the DSM-5 [7] does not include internet addiction or intensive internet use as an addictive disorder, nor does it include internet consumption in general within the “behavioral addictions”. Notwithstanding, there have been several important approaches to internet use in the last two decades. Thus, Tokunaga and Rains [8] refer to three great traditions such as: (a) problematic internet use (PIU) as an impulse control disorder (these authors have conceptualized it as a product of inadequate control over impulsive thoughts or feelings); (b) PIU as a substance dependence (this has been regarded as analogous to a chemical addiction and is defined as internet dependency); and (c) PIU as a deficit of relational and relationship-building resources (negative effects of this internet use in the person’s life).

This study has focused on this last approach to PIU, which involves emphasizing the possible dysfunctions entailed by the consumption of internet in the individual’s life [9,10,11]. According to this theoretical approach, PIU must be understood as a preference for online social interaction and for mood regulation through the internet, which increases the probability of presenting deficient self-regulation (either due to compulsive internet use or to excessive concern about it) and which leads to various negative consequences in the person’s life [9]. We must also consider that maladaptive cognitions about oneself and the world are considered part of the concept of PIU. These thoughts generate a feeling of inefficiency in offline social interactions and the possibility to overcome it through online relations [9,12]. In fact, it has been observed that online behavior carried out by adolescents with high social anxiety and low skills is mainly motivated by the desire to make friends [13]. There is also a tendency to think about online social interactions when the internet is not available, or to anticipate what will happen when one reconnects, which is a concern for the individual. To finalize this approach, we indicate that deficits in relationship-building resources lead to aversive moods, which makes people feel isolated and withdrawn [11,12,14].

Irrespective of the still controversial conceptualization of PIU, epidemiological studies report prevalence rates that demonstrate the clinical and social relevance of inappropriate internet use, especially when it entails risks and problems for a person [5,15,16,17,18]. Although it is difficult to talk about the general prevalence of PIU because of the many tools and ways of defining the construct [8,19], we note some studies in the international context such as that of Cha and Seo [20], who reported that almost 17% of adolescents present problematic use of the internet and/or the smartphone, and that up to 31% present risk of an addiction. In European studies, the prevalence reported is very variable: ranging from 4.4% of pathological internet use and 13.5% of maladaptive internet use (MIU) [21] up to between 14.3% and 54.9% [22]. In the Spanish context, Carbonell et al. (2012) found 6.1% of frequent problems due to internet use in a sample that included an age ranging from preadolescence to young people [23]. González–Cabrera et al. (2017) noted prevalences close to 4% in adolescents who presented PIU, and almost 40% presented occasional problematic use in the last seven months [4]. Other studies have found data with similar values to the ones presented [5,24]. However, other Spanish [25] authors suggest higher prevalences, indicating that up to 19.9% of Spanish adolescents present PIU, and, in another study with a very broad sample, 16.3% [26]. Finally, some studies have tried to establish groups of users according to adolescents’ profile of internet usage [27,28,29,30]. The results of these studies are not unanimous, although they often find profiles related to the ones proposed by Brandtzæg (2010) depending on the frequency of use, ranging from low use (sporadics), average use (lurkers/instrumental users/socializers) to high use (advanced users). In a Spanish study, Rial et al., (2015) replicated this taxonomy, finding four user profiles similar to the sporadics, socializers, instrumental, and advanced users of Brandtzæg [28], which they called first steppers, trainees, sensible users, and heavy users, respectively [30].

With regard to gender, some studies point out that PIU is higher in boys than in girls [21,31] whereas, in other investigations, it is higher in girls than in boys [5,18,24,26]. With regard to age, PIU seems to increase with age [32]. Studies with samples of adolescents suggest that post-compulsory students (late adolescence) present more PIU than the early courses (10–12 years) [26], although other authors find no age-related differences [24]. If young people are compared with middle aged people, the former have higher levels of PIU [18].

The problematic use that some adolescents make of the internet has also been shown and it has been reported to have a negative impact on health including changes in health habits (sleep, eating, physical activity, etc.) and their interference in the family, social, and academic life indicated in several studies [33,34]. Some variables associated with an increased risk of manifesting problematic internet-related behaviors have also been identified. Specifically, it has been associated with cyberbullying [24,35], with some personality traits, such as higher neuroticism and lower responsibility, openness to change, and agreeableness [16], as well as with loss of control, feelings of anger, stress symptoms, social isolation, family conflicts [33], and anxiety and depression [19].

In relation to the instruments that have evaluated PIU, there is a wide variety (cfr. Rial et al., 2016, [36] for a more thorough review of this and other internet-related constructs). In particular, instruments such as the Generalized and Problematic Internet Use Scale-2 (GPIUS2) [9,10,11], the OCS [14], and the PIUS [37] have been designed based on the above perspective of considering PIU as a relational resource deficit [8]. Of these three, the GPIUS2 [9,10] is the most complete, as it integrates the dimensions of uncontrolled internet use, use preoccupation, mood alteration, online efficacy, and negative outcomes. Neither the GPIUS2 nor any of the other instruments present the dimension of tolerance/withdrawal [8]. The GPIUS2 is one of the most widely used in relation to this approximation to PIU and it currently presents the limitation of a lack of cut-off points for the detection of problematic use [10]. Few studies have focused on establishing cut-off points, because the samples are generally not sufficiently large or representative [23,38,39]. In Spanish, the only known case of a tool that presents cut-off points is that of Gómez et al. [25]. However, their tool is far from the conceptualization of Caplan [9,11].

Health-related quality of life (HRQoL) is a complex construct for which there is no accepted definition [40]. Nevertheless, there is a clear consensus to not define it as an absence of disease or disorder but instead from a more holistic point of view that integrates physical, psychological, emotional, and social aspects. Moreover, well-being must be perceived by the subject him—or herself—and also by those around him or her [40,41,42]. HRQoL has been widely studied in adult population in numerous medical pathologies, and many instruments have been developed to measure it [43]. Recently, the European KIDSCREEN instruments, aimed at the evaluation of HRQoL in children and adolescents, were developed [41]. During childhood–adolescence, evidence has revealed that girls suffer a greater loss of physical and psychological HRQoL than do boys [44]. Some of the psychosocial problems that are beginning to be assessed in relation to HRQoL are those associated with violence. These include abuse [45], bullying [46], or cyberbullying [47].

Therefore, at present, PIU has hardly been related to HRQoL. Although there is evidence that adaptive internet use has positive effects on children’s psychological well-being [48,49], currently research focuses on problems related to problematic, addictive, or maladaptive use related to the loss of HRQoL or of psychological well-being. Despite the evidence that has been found, there is currently great heterogeneity in the way we define and evaluate the two constructs. Thus, lower HRQoL has been reported in adolescents and young adults with PIU than in those without PIU [50,51]. This relationship between inadequate internet use and poorer HRQoL has also been observed in primary education students [52]. There is evidence of the inverse relationship between PIU and HRQoL [53] and between internet addiction and HRQoL [54]. Other studies have explored the use of social media, and online games are related to poorer psychological well-being, especially among girls [55]. From Caplan’s approach to PIU, it could be considered that online social preference and mood regulation through the internet generate social interactions that provoke feelings of emotional loneliness even if the individual apparently has social support, because this kind of communication lacks the sensory richness that human beings need [56,57]. This is also supported by studies indicating the two-way relationship between emotional loneliness and PIU, especially when the use of social media is predominant [58,59].

Considering the above, the present study has as its main objectives: (1) to analyze the profiles related to PIU and (2) to analyze the relationship between problematic use and HRQoL in a broad sample of adolescents. In addition, we had the following secondary objectives: (1) to analyze sex and age in relation to PIU; (2) to analyze the prevalence of PIU in adolescents; (3) to compare HRQoL as a function of the profile of PIU; and (4) to establish a cut-off point for PIU.

The study had the following hypotheses: (H1) there will be significant and inverse relationships between PIU and HRQoL; (H2) the greater the level of PIU, especially if it is severe, the worse will be HRQoL [54].

## 2. Materials and Methods

### 2.1. Design and Participants

An analytical and cross-sectional study was carried out in an autonomous community in northern Spain. The sample comprised 12,285 participants of whom 49.1% were boys (*n* = 6032) and 50.3% girls (*n* = 6181); 72 participants did not provide this datum (0.6%). The sample obtained in the study is representative of students enrolled in public and concerted centers. With a 99% confidence level, a sampling error of 1% for a population variance of 0.50, the representative sample comprises 11,853 participants. In addition, sample selection was random, taking into account the number of students of each stage, the proportionality of centers in each location, and the type of center. Mean age and standard deviation was 14.69 ± 1.73 with a range of 11–18 years. Concerning age, 40% (*n* = 4912) of the sample was between 11–13 years old, 37.3% (*n* = 4580) between 14–15 years, and 22.45% (*n* = 2766) between 16–18 years, with 27 (0.2%) participants who did not provide this datum. Fifty-nine schools participated, and more than 90% of the students of the centers were recruited. Concerning ownership, 16 schools were private schools (*n* = 3381; 27.25%) and 43 were public schools (*n* = 8904; 72.5%). The average socioeconomic and cultural index (ISEC) of the schools of the sample is similar to the mean ISEC of the autonomous community to which they belong, according to data obtained from the diagnostic assessments in the 2009–2014 period.

### 2.2. Instruments

The participants provided information about demographic variables such as sex, grade, school, and age. For the analysis of the variables under study, the following instruments were used in relation to the previous five months (start of the course).

Spanish version of the GPIUS2 [9,10]. It has 15 items divided into five factors: preference for online social interaction (e.g., “I prefer to interact with other people through the internet rather than communicating face to face”), mood regulation (e.g., “I’ve used the internet to talk to others when I've felt lonely”), Negative consequences (e.g., “My use of the internet has hindered the control of my life”), cognitive concern (e.g., “I would feel lost if I couldn't connect to the internet”), and compulsive use (e.g., “When I'm not on the internet, it’s hard to resist the urge to connect”). Agreement with the items is rated on a six-point Likert scale ranging from 1 (completely disagree) to 6 (completely agree). The reliability ratings can be seen in Table 5.

For the evaluation of HRQoL, we used the Spanish version of the KIDSCREEN-27 [41] for children and adolescents aged 8 to 18 years. This version evaluates five dimensions through 27 items: physical well-being (e.g., “Have you felt well and fit?”), psychological well-being (e.g., “Have you felt sad?”), autonomy and relationship with parents (e.g., “Have your parents had enough time for you?”), friends and social support (e.g., “Have you and your friends helped each other?”), and school environment (e.g., “Have you done well in school?”). The development of the KIDSCREEN was based on the probabilistic partial credit model (PCM), which belongs to the family of Rasch models. PCM tries to explain the actual behavior of the respondents in the testing situation by the estimated the person parameter and the location of the item-response-category-thresholds. The PCM assumes all items of a scale to be indicators of a single unidimensional latent trait [41]. For the KIDSCREEN-27, the mean scores varied around 50 (*SD* = 10) due to *T*-value standardization. There are standardized data for the Spanish infant-juvenile population. The reliability indexes can be seen in Table 5.

### 2.3. Procedure

The application of the questionnaires to the students of the different classrooms was supervised by the tutor, aided by the Guidance Department and the school direction. A rigorous procedure was established for data collection through the Survey Monkey^®^ online platform, using the computer labs of each school. We emphasized the importance of responding sincerely, individually, and reflectively, but without taking too much time on each item. The average response time for the questionnaire battery was approximately 12 minutes. Collaboration was voluntary, anonymous, and disinterested.

The study was carried out with the authorization of the participants, the schools, and the political-educational institution of the autonomous community. Through the official communication channels with the families, the schools sent a passive consent form that informed the legal tutors about the purpose of the study and its characteristics, its promoters and their right not to participate. Those parents/tutors who did not wish to allow participation returned the signed consent refusing participation. This occurred in less than 1% of the sample. The project was approved by the Ethics Committee of the Principality of Asturias (Ref.59/17), and the juvenile Prosecutor’s Office was informed, following the legal regulations in force. There were no exclusion criteria, except for the refusal to participate by the legal guardians or by the students themselves.

### 2.4. Data Analysis

Statistical analyses were carried out using the Statistical Package for the Social Sciences (SPSS) [60], the R software program [61], and the Pysch [62] and Mclust [63] package. Firstly, to determine the internal consistency of the instruments used, Cronbach alphas [64] and omega coefficients [65] were estimated. Then, we checked the assumptions of normality of the target variables of the study (Shapiro-Wilks statistic), as well as the homogeneity of variances to compare the groups (Levene test).

The following analyses were carried out on the GPIUS2 scores: (1) Student’s *t* for dependent and independent samples to determine possible sex differences in the GPIUS2 dimensions; (2) ANOVA with post-hoc Bonferroni comparisons to determine age group differences; (3) Latent Profile Analysis (LPA) to identify profiles of internet use, taking into account the five dimensions of the GPIUS2; for this purpose, exploratorily, we compared different solutions according to the Bayesian criterion information (BIC) and the Akaike information criterion (AIC). Latent profile analysis (LPA) tries to identify clusters of individuals based on responses to a series of continuous variables. LPA assumes that there are unobserved latent profiles that generate patterns of responses on indicator items. The resulting profiles can then inform behavioral or other important outcome differences and help explain inconsistencies in prior research [66]; (4) Chi-square analysis and analysis of the adjusted standardized residuals to identify the distribution of sex and age in the profiles; (5) receiving operating characteristic (ROC) curve analysis, using the “severe problematic use” profile as the gold standard to distinguish between two binary elements (i.e., severe PIU and non-problematic internet use). The ROC curve analysis identified the cut-off points based on their sensitivity (i.e., the percentage of true positives among problematic users) and their specificity (i.e., the percentage of true negatives among users with non-problematic use). We also analyzed the GPIUS2’s likelihood of making a correct diagnosis through positive (PV+) and negative predictive values (PV−), and lastly, its overall accuracy or efficiency (ACC), indicating the proportion of valid results among all the results.

To determine the relation between the GPIU and its effects on HRQoL, the following analyses were carried out; (1) partial Pearson correlations for both sexes (with age as control variable) between the five GPIUS2 dimensions and the five KIDSCREEN dimensions; (2) ANOVA with post-hoc Bonferroni comparisons of the profiles obtained with LPA (independent variable) and the scores obtained in the five KIDSCREEN dimensions to analyze HRQoL differences as a function of PIU. A value of less than *p* = 0.05 was considered significant.

## 3. Results

### 3.1. Differences in PIU as a Function of Sex and Age

Sex and age differences in problematic use are shown in Table 1. Girls obtained significantly higher scores in the total GPIU than boys, although the size of the effect was small *(t* = −7.23, *p* < 0.001, *d* = 0.015). More specifically, girls showed significantly higher scores in the variables: mood regulation, cognitive preoccupation, and compulsive use. Age-based analyses revealed significant differences in GPIUS2 (*F* (2,12280) = 127.12, *p* < 0.001, *η*_p_^2^ = 0.020) and its five dimensions. In particular, the group of 11–13-year-olds obtained lower scores in all the dimensions, whereas the group of 14–15-year-olds obtained the highest scores.

### 3.2. Profiles of Internet Use and Effects on Quality of Life

To compare the scores obtained in the GPIUS2 and its five dimensions and their relationship with HRQoL, we performed LPA. Table 2 shows the values obtained for the different models. Inspection of the fit indices suggests a four-profile solution as the best fit, as the fit indices began to level off after additional profiles. The Bootstrap Likelihood Ratio Test provided a significant *p*-Value, indicating a good fit for the four-profile solution, which ceased to be significant when adding more profiles. This solution offered high values of entropy and the average posterior probabilities for the membership in each latent profile were high (from 0.85 for “mood regulation use” to 0.96 for the “severe problematic problem”).

The mean (standardized) scores obtained in the dimensions of the GPIUS2 are shown in Figure 1, and their respective scores in Table 3.

The profile that comprised the largest number of participants was the “non-problematic” profile, characterized by scores below the average in all five GPIUS2 dimensions. Next, two profiles emerged with a similar number of participants, but with different usage patterns: the “mood regulator” profile showed a high level of usage for mood regulation and high online preference, but it obtained scores below the average in the rest of the dimensions. The “problematic internet use” profile showed above average values in all the GPIUS2 dimensions, but much lower scores than the last profile, “severe problematic-compulsive use”, which obtained the highest scores in all GPIUS2 dimensions, with particularly high scores in negative consequences, cognitive preoccupation, and compulsive use.

Table 4 shows the profiles according to sex and age. Chi-square analyses yielded significant differences for both variables. The analysis of the standardized residuals indicated a higher number of boys in the non-problematic users group and a higher number of girls in the problematic users and severe problematic users groups. Regarding age, the standardized residuals indicated a smaller number of 11–13-year-olds and a larger number of 16–17-year-olds in the three profiles related to dysfunctional use. However, more 11–13-year-olds and fewer 14–15- and 16–17-year-olds than expected were found in the non-problematic use profile.

### 3.3. Relationship between GPIU and HRQoL

Next, we examined the relation between the GPIU and HRQoL. The correlations between the five GPIUS2 dimensions and the five HRQoL dimensions, controlling age for both sexes, are presented in Table 5. In the sample as a whole, significant negative correlations were observed between the GPIUS2 dimensions and KIDSCREEN dimensions, with values ranging between *r* = −0.101 (cognitive preoccupation and peer relations) and *r* = −0.334 (mood regulation and psychological well-being). As can be seen in Table 5, the partial correlations were weaker than those obtained with the sample as a whole, particularly in the case of boys in the parents relations and autonomy dimension and online preference (*r* = −0.069), whereas girls obtained higher correlations between the School environment dimension and the five GPIUS2 dimensions.

Then, ANOVAs were carried out to compare the scores in the five HRQoL dimensions as a function of the LPA profiles of PIU. The results can be seen in Table 6.

### 3.4. GPIUS-2 Cut-Off Point

In order to create a cut-off point for the GPIUS2, a ROC curve analysis was performed. The result of this analysis showed an area under the curve (AUC) of 0.975, indicating a suitable potential of the GPIUS2 to discriminate between problematic and non-problematic internet users (see Figure 2).

Table 7 shows the sensitivity, specificity, PV+, PV−, and ACC of the GPIUS2 and its different cut-off points, calculated using the profile of “severe problematic users” as the gold standard. The analysis led to a number of possibilities based on the authors’ decisions, in which a specific approach was prioritized due to the low prevalence of the problem and the primary interest of avoiding the generation of false positives. Hence, the value of 52 was proposed as a cut-off point to differentiate users with PIU from those who do not have this problem. At this point, a specificity of 96% and a sensitivity of 93% were obtained. PV− was higher than 99%, so less than 1% of those who were not classified as problematic users were misdiagnosed. Finally, the ACC for the cut-off point 52 was 96%.

## 4. Discussion

This paper contributes to our knowledge of adolescents’ PIU and HRQoL. It also delves into a little-explored reality by analyzing differences in HRQoL as a function of the level of severity associated with PIU and it provides a cut-off point for a widely used tool in this topic.

Although it has used a conceptualization of PIU that considers it an artifact of relational and relationship-building resource deficits, the data obtained are consistent with those found in other studies. Approximately 5% of the study sample presented severe problems, which coincides with previous studies in the Spanish context [4,5,23,24]. If the problematic use and the severe problematic use profiles are considered jointly, the prevalence is about 23%, which is consistent with other studies in the national [25,26] or in European contexts [22]. In relation to sex and PIU, the data show that there are more problems in girls than in boys, as had been reported in previous studies [5,18,24,26,55]. However, this is a controversial issue, as other studies indicate the opposite [21,31]. Part of these differences may be due to the conceptualization of PIU, especially if the focus is on internet addiction or on specific problematic internet uses such as online gambling [18]. Another possible explanation is related to the evaluation of PIU in girls, particularly focused on the use of social media [55,67]. Boys generally present higher levels of other internet uses such as online video games [68]. The data also suggest that there is lower prevalence of problematic use during early adolescence (10–12 years), in line with other studies [26,52,55].

Various studies have linked the inappropriate use of the internet with psychological problems such as psychological stress and symptoms of mental disorders [38], behavioral problems, hyperactivity, impact on daily life activities, depression, and poor physical health [69]. Other authors have pointed out that inadequate use of the internet is related to low scores on the HRQoL, as well as to lower self-perceived social support and more friends known only through the internet [52,70]. In this same line, the data from this study provide evidence of the relationship between HRQoL and PIU, from the perspective of Caplan [9,11], which placed special emphasis on mood regulation through the internet and the preference for online relationships. However, the data are also convergent with other approaches [47,50,51,52,53]. The most important (inverse and significant) relationships were found between mood regulation and negative consequences and psychological well-being. The loss of psychological well-being in those adolescents with problematic use and, especially, with severe problematic use, is very acute (a bit more than one total standard deviation). These data confirm the first and second hypotheses (H1 significant and inverse relationships between PIU and HRQoL; and H2 the greater the level of PIU, especially if it is severe, the worse will be HRQoL) of the manuscript. It is possible that excess emotion regulation and online social preference are associated with high levels of emotional loneliness, which would influence psychological well-being [56].

LPA identified four subgroups, three of which were expected based on the literature [5,28,30]: non-problematic use, problematic use, and severe problematic use. However, a profile emerged that was called “mood regulator” which presents high scores in the dimension of mood regulation and online preference and low scores in the remaining dimensions. This is evidence of the interest in applying new statistical techniques to already established knowledge [66]. The characteristics of the “mood regulator” profile use match the theoretical model proposed by Caplan [9,11] but without problems due to negative consequences. As shown in the results, the impact on the KIDSCREEN-27 dimensions increases as the use of the internet becomes more problematic. Inspection of the affected dimensions in the mood regulator profile shows that this style could be prior to—or derive in—more problematic internet usage, depending on other factors such as connection time, gratifying experiences, and its influence in the academic-work, family, or social areas. In turn, these results are consistent with other studies that identify different types of users [28,30], although they consider additional variables. In particular, the no-problem group would be equivalent to the first steppers or sporadic groups, the group of mood regulator users could resemble the group of entertainment users, whereas the problematic user or severe problematic user profiles would be related to the heavy users or advanced user groups. This expands a future line of research, as prevention and intervention programs could focus on these profiles and analyze their evolution over time and through motives for change.

This work obtained a cut-off point (52), which represents a unique contribution to the literature. There are few works in this line that establish cut-off points to differentiate people with or without problems in internet contexts [25,68]. We sought a balance between all the indicators, while being especially cautious when making decisions because of the diagnostic implications that are associated with labeling [71,72]. Thus, a high level of specificity of the test prevailed, as a specific test is useful when it is positive, and this is especially important in problems with low prevalence, such as severe problematic use (about 5%). In similar vein, the chosen cut-off point has an especially high negative predictive value (99.6%), which means that almost 100% of the subjects who were classified as negative do not, in fact, have a severe problem. We chose this conservative usage to favor the diagnosis of clearly severe cases and to avoid over-diagnosis.

However, the study is not exempt from limitations. First, the results are based on self-reports with the entailed response bias. In the future, additional measures are proposed such as sociograms or parent/teacher/peer reports. This is a cross-sectional study, which does not allow establishing causal relationships between the study dimensions, so, in the future, studies should propose longitudinal designs. Although the sample is broad and representative of a Spanish region, we must be cautious when extrapolating the results to other cultural contexts. The cut-off point was not estimated by any external measure (e.g., another validated PIU screening tool and/or practitioner’s diagnosis/judgement), although a similar methodology has been used elsewhere in the scientific literature [68]. Future studies will be necessary to re-examine these results and use other external measures. Although this is not a limitation in itself, it must be taken into account that PIU was conceptualized in a certain way, following the model of Caplan [9,11], not contemplating other theoretical formulations, and this makes the comparison of data with other studies more difficult. It is also possible that “mood regulating” users are more than entertainment users, as too much mood regulation through external activities may be linked to psychopathology, and this was not evaluated. We also note that a multilevel analysis could have been considered, taking into account the possible variability between classes and schools, but this was not possible because the data were not collected by classes, but only by schools due to the planning of the study with the educational authorities. It would also be advisable in the future to include other instruments, especially those with validations in many languages, to facilitate the comparison of results and to compare theoretical proposals of PIU. Constructs related to internet gaming disorder or online gambling could also be introduced as specific problems in order to analyze their influence on HRQoL and other constructs related to psychological well-being.

The study has several practical implications for professionals in the educational and clinical field. The GPIUS2 can be administered quickly as of 10 years of age and can provide an idea of whether or not the person presents problematic use problems. The evidence of more PIU in girls and in older adolescents reveals a reality that should be considered in order to adapt the different interventions in this problem. Likewise, the cut-off point reached in this study clearly establishes the score above which we can consider an adolescent to present PIU, with the practical, educational, and clinical implications that this entails. Also, the loss of HRQoL in adolescence without a medical cause may be associated with psychosocial problems such as those explored in this study and can be considered from the perspective of health professionals (pediatricians, psychologists, etc.). Similarly, prevention strategies and programs related to internet problems should increasingly focus on the problematic uses of the internet or even, as has been shown, its use as a mood-regulator. It is expected that adequate prevention will reduce these problems and help adolescents to make a healthier and more positive use of the internet.

## 5. Conclusions

Four internet profiles were found in terms of adolescent usage habits. There is a negative relationship between PIU and HRQoL. In particular, adolescents with severe PIU showed a greater impact on their HRQoL, although those who used the internet as a mood-regulator and those who presented problematic use also showed impaired HRQoL. In addition, the score of 52 in the GPIUS is proposed as a cut-off point to distinguish users with severe PIU.

## Figures and Tables

**Figure 1 ijerph-16-03877-f001:**
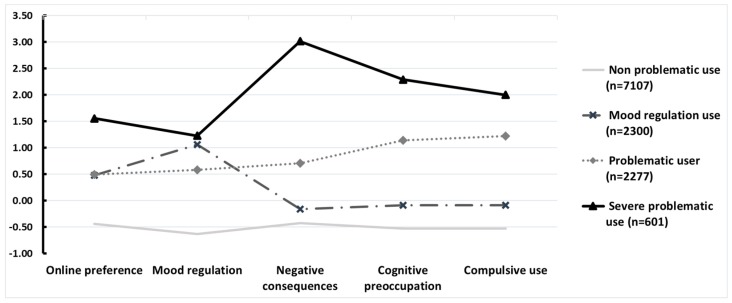
Profiles of problematic internet use (standardized scores).

**Figure 2 ijerph-16-03877-f002:**
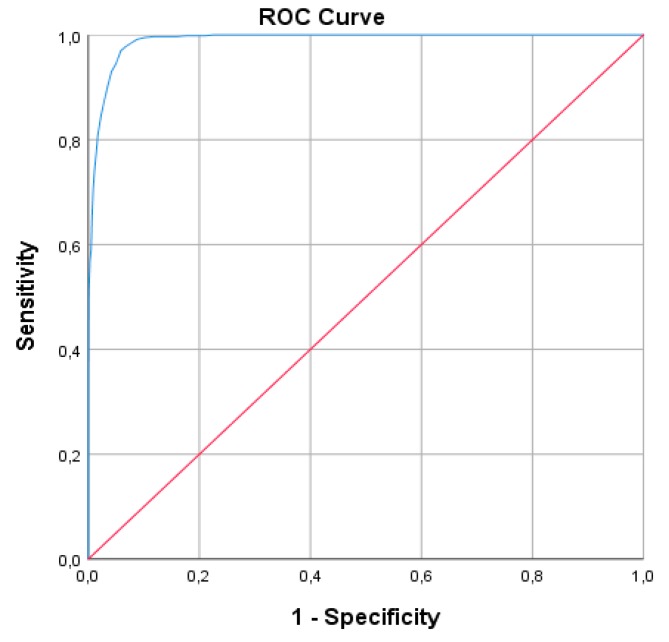
The receiver operating characteristic (ROC) curve of the Spanish version of GPIUS2.

**Table 1 ijerph-16-03877-t001:** Differences as a function of the variables sex and age groups (11–13 years, 14–15 years, and 16–18 years) in the Generalized and Problematic Internet Use Scale-2 (GPIUS2) dimensions.

	Boys	Girls	*t* (*p*)	*d*	11–13 Years ^a^	14–15 Years ^b^	16–18 Years ^c^	*F* (*p*)	*η* ^2^	Post Hoc
	*M (SD)*	*M (SD)*			*M (SD)*	*M (SD)*	*M (SD)*			
Online Preference	5.60 (3.33)	5.56 (3.35)	0.696 (0.486)	0.01	5.23 (3.15)	5.73 (3.37)	5.71 (3.43)	27.99 (0.000)	0.005	a < b; a < c
Mood Regulation	7.81 (4.48)	8.39 (4.59)	7.01 (0.001)	0.13	7.19 (4.39)	8.23 (4.58)	8.75 (4.49)	119.04 (0.000)	0.019	a < b < c
Negative Consequences	4.55 (2.67)	4.63 (2.76)	1.61 (0.107)	0.02	4.16 (2.39)	4.63 (2.74)	4.91 (2.88)	77.20 (0.000)	0.012	a < b < c
Cognitive Preoccupation	5.28 (3.15)	5.85 (3.44)	8.61 (0.000)	0.17	4.92 (3.01)	5.73 (3.35)	5.96 (3.44)	92.67 (0.000)	0.015	a < b < c
Compulsive Use	6.28 (3.73)	6.88 (4.04)	7.26 (0.001)	0.15	6.01 (3.69)	6.69 (3.96)	6.96 (3.95)	61.31 (0.000)	0.010	a < b < c

Note: Boys (*n* = 6032); girls (*n* = 6181); 11–13 years (*n* = 4912), 14–15 years (*n* = 4580); 16–18 years (*n* = 2766), *t* = Student’s *t*; *p* = significance; *d* = Cohen’s *d*; *F* = Fishers *F*; *η*^2^ = partial eta squared; Post hoc = Bonferroni’s post hoc comparison.

**Table 2 ijerph-16-03877-t002:** Fit of the 2–5 profile models based on the GPIUS2 dimensions.

Profile	AIC	BIC	Sample Size Adjusted BIC	LL	*p*-Value for BLRTS	Entropy	*n* Per Profile				
							1	2	3	4	5
2	153,323.079	153,441.738	153,390.891	76,645.540	0.001	0.98	9645	2640			
3	146,752.960	146,916.115	146,846.201	73,354.480	0.001	0.95	9383	3071	831		
**4**	**144,715.491**	**144,923.143**	**144,834.162**	**72,329.745**	**0.001**	**0.90**	**7107**	**2300**	**2277**	**609**	
5	144,730.976	144,983.124	144,875.076	72,331.488	1.0	0.66	-	-	-	-	-

AIC = Akaike information criterion; BIC = Bayesian information criterion; LL = likelihood logarithm; BLRT = bootstrapped likelihood ratio test sequential. The selected model (four-profile model) is shown in boldface.

**Table 3 ijerph-16-03877-t003:** Differences in the dimensions of the GPIUS-2 as a function internet usage profile (*n* = 12,258).

GPIUS Profile	Online Preference		Mood Regulation		Negative Consequences		Cognitive Preoccupation		Compulsive Use	
	*M (SD)*		*M (SD)*		*M (SD)*		*M (SD)*		*M (SD)*	
No problem ^a^	4.10	1.84	5.22	2.51	3.43	1.02	3.81	1.35	4.52	2.15
Mood regulation use ^b^	7.17	3.63	12.92	2.81	4.16	1.60	5.27	1.94	6.23	2.52
Problematic use ^c^	7.22	3.49	10.74	3.96	6.50	2.45	9.34	2.80	11.33	2.65
Severe problematic use ^d^	10.77	4.29	13.67	3.63	12.77	2.85	13.16	3.26	14.37	2.90
*F (df)* η^2^	*F* (4,12281) = 1919.67 ***;		*F* (4,12281) = 5646.19 ***;		*F* (4,12281) = 7925.33 ***;		*F* (4,12281) = 7952.08;		*F* (4,12281) = 7090.52 ***;	
	*η*^2^ = 0.319;		*η*^2^ = 0.580;		*η*^2^ = 0.648;		*η*^2^ = 0.660 ***;		*η*^2^ = 0.634;	
Post hoc Bonferroni (Cohen’s *d*)	a < b (1.27);		a < b (2.98);		a < b (0.61);		a < b (0.80);		a < b (0.80);	
	a < c (1.33);		a < c (1.88);		a < c (2.05);		a < c (2.16);		a < c (2.16);	
	a < d (3.12);		a < d (3.23);		a < d (7.40);		a < d (3.74);		a < d (3.74);	
			b > c (0.64);		b < c (1.10)		b < c (1.69);		b < c (1.69);	
	b < d (0.95);		b < d (0.25);		b < d (4.47);		b < d (3.46);		b < d (3.46);	
	c < d (0.97);		c < d (0.75);		c < d (2.40);		c < d (1.32);		c < d (1.32);	

Note: M = arithmetic mean; SD = standard deviation; *p* = significance; *F* = Fishers *F*; df = degrees of freedom; *η*^2^ = eta squared; *** = *p* < 0.001.

**Table 4 ijerph-16-03877-t004:** Sex and age distribution in the GPIU profiles.

GPIU Profile	Male (*n* = 6032)	Female (*n* = 6181)	χ^2^ (3)	11–13 years (*n* = 3606)	14–15 years (*n* = 4516)	16–17 years (*n* = 4160)	χ^2^ (6)
	*f* (%) ^a^	*f* (%)		*f* (%) ^b^	*f* (%)	*f* (%)	
Non problematic use (*n* = 7107)	3643 (60.4) *	3424 (55.4) **	42.96 (<0.001)	2448 (67.90) *	2503 (55.4) **	2156 (51.8) **	255.80 (<0.001)
Mood regulation use (*n* = 2300)	1123 (18.6)	1167 (18.9)		541 (15.0) **	895 (19.8) *	864 (20.8) *	
Problematic use (*n* = 2277)	1007 (16.7) **	1250 (20.2) *		495 (13.7) **	883 (19.5) *	899 (21.6) *	
Severe problematic use (*n* = 601)	259 (4.3) **	272 (3.1) *		122 (3.4) **	236 (4.1)	207 (5.8) *	

Note: *f* = frequency; % = percentage; χ^2^ (df); Chi Squared (degrees of freedom); ^a^ percentage indicated over each column (sex); ^b^ percentage indicated over each column (age group). * Adjusted standardized residuals > 1.96. ** Adjusted standardized residuals < −1.96.

**Table 5 ijerph-16-03877-t005:** Partial correlations (controlling for age) between the GPIUS2 dimensions and the five KIDSCREEN-27 dimensions in boys and girls. Means and standard deviations are included for the study sample (*n* = 12,285).

		1.	2.	3.	4.	5.	6.	7.	8.	9.	10.	*M (SD)*	α	ω
GPIUS−2 Dimensions	1. Online preference	—	0.468	0.392	0.412	0.350	−0.207	−0.292	−0.213	−0.218	−0.221	5.58 (3.34)	0.85	0.85
	2. Mood regulation	0.471	—	0.367	0.456	0.404	−0.214	−0.358	−0.262	−0.185	−0.271	8.10 (4.54)	0.82	0.83
	3. Negative consequences	0.432	0.365	—	0.562	0.593	−0.206	−0.293	−0.241	−0.205	−0.260	4.59 (2.71)	0.76	0.77
	4. Cognitive preoccupation	0.460	0.451	0.609	—	0.734	−0.172	−0.228	−0.175	−0.089	−0.224	5.57 (3.31)	0.79	0.79
	5. Compulsive use	0.422	0.419	0.611	0.702	—	−0.190	−0.229	−0.188	−0.091	−0.234	6.58 (3.90)	0.82	0.84
KIDSCREEN Dimensions	6. Physical well−being	−0.183	−0.144	−0.169	−0.132	−0.175	—	0.453	0.376	0.321	0.415	49.14 (12.00)	0.87	0.88
	7. Psychological well−being	−0.251	−0.301	−0.291	−0.250	−0.251	0.401	—	0.546	0.504	0.544	48.53 (10.13)	0.86	0.86
	8. Parents relations and autonomy	−0.169	−0.204	−0.219	−0.181	−0.212	0.358	0.480	—	0.440	0.490	52.83 (10.91)	0.84	0.84
	9. Social support and peers	−0.189	−0.131	−0.191	−0.121	−0.143	0.367	0.490	0.475	—	0.386	54.09 (10.34)	0.85	0.85
	10. School environment	−0.150	−0.183	−0.202	−0.179	−0.207	0.299	0.425	0.459	0.373	—	49.73 (10.32)	0.80	0.81

Note: The correlations for boys are shown below the diagonal, and for girls above it. All correlations were significant at the level *p* < 0.001. M = arithmetic mean; *SD* = standard deviation; α = Cronbach’s alpha; ω = McDonald’s omega.

**Table 6 ijerph-16-03877-t006:** Differences in the KIDSCREEN-27 dimensions as a function of the internet use profile (*n* = 12,258).

	Phy-wb	Psy-wb	Pr&A	SS&P	SchEn
GPIU Profile	*M (SD)*	*M (SD)*	*M (SD)*	*M (SD)*	*M (SD)*
No problem ^a^	51.30 (11.89)	51.36 (9.89)	55.19 (10.88)	55.64 (10.15)	52.02 (10.42)
Mood regulation use ^b^	47.51 (11.13)	45.54 (8.90)	50.54 (10.13)	52.61 (9.87)	47.66 (9.07)
Problematic use ^c^	45.76 (11.12)	44.79 (8.69)	49.28 (9.66)	52.01 (10.04)	46.32 (8.77)
Severe problematic use ^d^	42.61 (13.24)	40.64 (9.46)	47.24 (11.23)	49.47 (11.95)	43.51 (11.21)
*F(df) η* ^2^	*F* (3,12226) = 221.22 ***;	*F* (3,12144) = 534.34 ***;	*F* (3,12173) = 293.87 ***;	*F* (3,12263) = 143.94 ***;	*F* (3,12236) = 326.01 ***;
	*η*^2^ = 0.051;	*η*^2^ = 0.117;	*η*^2^ = 0.068;	*η*^2^ = 0.034;	*η*^2^ = 0.074;
Post hoc Bonferroni *(Cohen’s d)*	a > b (0.33);	a > b (0.62);	a > b (0.44);	a > b (0.30);	a > b (0.43);
	a > c (0.47);	a > c (0.71);	a > c (0.57);	a > c (0.36);	a > c (0.57);
	a > d (0.72);	a > d (1.10);	a > d (0.71);	a > d (0.60);	a > d (0.81);
	b > c (0.16);				b > c (0.15);
	b > d (0.40);	b > d (0.53);	b > d (0.30);	b > d (0.31);	b > d (0.43);
	c > d (0.27);	c > d (0.46)	c > d (0.20)	c > d (0.25)	c > d (0.30)

Note: Phy-wb = physical well-being; Psy-wb = psychological well-being; Pr&A = parents relations and autonomy; SS&P = social support and Peers; SchEn= school environment; M = arithmetic mean; SD = standard deviation; *p* = significance; *F* = Fisher’s *F*; *η*^2^ = eta squared. df = degrees of freedom; *** = *p* < 0.001.

**Table 7 ijerph-16-03877-t007:** Cut-off point for GPIUS-2 based on the “severe problematic users” profile obtained through LPA.

Cut-Point	TP	TN	FP	FN	Sens.	Spec.	PV+	PV−	ACC
46	598	10,491	1193	3	99.50%	89.79%	33.39%	99.97%	90.26%
47	596	10,654	1030	5	99.17%	91.18%	36.65%	99.95%	91.58%
48	592	10,772	912	9	98.50%	92.19%	39.36%	99.92%	92.50%
49	588	10,887	787	13	97.84%	93.26%	42.76%	99.88%	93.41%
50	583	10,998	686	18	97.00%	94.13%	45.94%	99.84%	94.27%
51	568	11,101	583	33	94.51%	95.01%	49.35%	99.70%	94.99%
52	559	11,196	488	42	93.01%	95.82%	53.39%	99.63%	95.69%
53	542	11,279	405	59	90.18%	96.53%	57.23%	99.48%	96.22%
54	525	11,350	334	76	87.35%	97.14%	61.12%	99.33%	96.66%
55	510	11,411	273	91	84.86%	97.66%	65.13%	99.21%	97.04%
56	486	11,484	200	115	80.87%	98.29%	70.85%	99.01%	97.44%
57	465	11,520	136	164	73.93%	98.83%	77.37%	98.60%	97.56%

Note: TP = True Positive; TN = True Negative; FP = False Positive; FN = False Negative; Sens= Sensitivity; Spec= Specificity; PV+ = Positive Predictive Value; PV− = Negative Predictive Value; ACC = Accuracy.

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
