# Peer review of "Profiles of Problematic Internet Use and Its Impact on Adolescents’ Health-Related Quality of Life"

_ijerph, 2019, doi:10.3390/ijerph16203877_

Round 1

Reviewer 1 Report

Re: ijerph-597427, Profiles of problematic internet use and its impact on adolescents'
health-related quality of life 

        The relationship between problematic internet use and poor heal-realted quality of life (HR-QoL) were tested in this study. There are some potential strength: the sample size is large, the aim is clear, and the

My biggest concern is on what the extra scientific contribution can be brought by the latent profile analysis approach in this topic. It seems that poor HR-QoL and higher score of the original tool (Generalized and Problematic Internet Use Scale-2, GPISU2), is a very intuitive. The finding, or scientific contribution with or without the profile approach, may be very similar. Why this approach is employed? What valuable new finding can be found in the result? What new clinical implication can be bring by this approach? These important questions should be clarified, before evaluate this study. The hypothesis and result should be revised extensively, the most analysis should be able to corresponding to a specific hypothesis. I am worry about the statement that the “sample is representative” here. The sample may be very representive to the study site (or region). However, I believe the finding of this study is not only apply to this region. The real population applying the finding, is beyond the region. A bar-chart is recommended for the post-hoc analysis. The result of pairwise comparison could be a difficult to have a clear and concise impression.

Author Response

REVIEWER COMMENTS FOR THE AUTHOR

Reviewer #1:

Reviewer #1: The relationship between problematic internet use and poor health-related quality of life (HR-QoL) were tested in this study. There are some potential strength: the sample size is large, the aim is clear, and the…

Authors Thank you for your general appraisal of the manuscript. We think that the rest of your commentary was left unfinished by mistake.

Reviewer #1: My biggest concern is on what the extra scientific contribution can be brought by the latent profile analysis approach in this topic. It seems that poor HR-QoL and higher score of the original tool (Generalized and Problematic Internet Use Scale-2, GPISU2), is a very intuitive. The finding, or scientific contribution with or without the profile approach, may be very similar. Why this approach is employed? What valuable new finding can be found in the result?

Authors Certainly, science is moving forward thanks to the intuition of many researchers and the empirical verification of those results. The inverse relationship between problematic Internet Use and Health-Related Quality of Life may seem obvious, but this should be subject to analysis with the necessary rigor.

Latent Profile Analysis (LPA) tries to identify clusters of individuals (i.e., latent profiles) based on responses to a series of continuous variables (i.e., indicators). LPA assumes that there are unobserved latent profiles that generate patterns of responses on indicator items. The resulting profiles can then inform behavioral or other important outcome differences and help explain inconsistencies in prior research. In addition, LPA provides additional knowledge above and beyond traditional approaches such as regression analysis, as it can model relationships among three or more variables that are more difficult to interpret using regression analysis (Zyphur, 2009).

Zyphur, M. J. (2009). When mindsets collide: Switching analytical mindsets to advance organizational science. Academy of Management Review, 34, 677-688.

In this study, LPA is relevant. Initially, it had been hypothesized that there would be three profiles (according to previous studies): non-problematic use, problematic use and severe problematic use. These nomenclatures may vary according to the authors (Brandtzæg, 2010; Rial et al., 2015).

However, thanks to the LPA, a profile emerges that has been called a "mood regulator”. This profile presents high scores in the mood regulation and online preference dimensions and low scores in the remaining dimensions. This profile makes sense theoretically within the model proposed by Caplan (2002, 2010). This is evidence of the interest in applying new statistical techniques to already established knowledge.

Brandtzæg, P.B. Towards a unified Media-User Typology (MUT): A meta-analysis and review of the research literature on media-user typologies. Comput. Human Behav. 2010, 26, 940–956.

Rial, A.; Gómez, P.; Picón, E.; Braña, T.; Varela, J. Identification and Characterization of Adolescent Internet User’s Profiles. Span. J. Psychol. 2015, 18, 1–10.

Reviewer #1: What new clinical implication can be bring by this approach? These important questions should be clarified, before evaluate this study. The hypothesis and result should be revised extensively, the most analysis should be able to corresponding to a specific hypothesis.

Authors This work provides a cut-off point (52) for the GPIUS-2 which represents a singular contribution to the literature. There are few works in this line that establish cut-off points to differentiate people with or without problems in Internet contexts. The most approximate study to ours was carried out by Gómez, Rial, Braña, Golpe and Varela (2017), but it does not focus on the GPIUS2 questionnaire (nor on the Caplan model on Problematic Use of the Internet). We believe that the contribution of this study is to have a cut-off point (a first approximation) that allows a quick screening on possible problematic uses is a novel finding.

This same procedure has been carried out on other specific Internet problems (such as the Internet Gaming Disorder by Fuster, Carbonell, Pontes and Griffiths, 2016).

In the current version of the study, there are four hypotheses

The study had the following hypotheses: a) the prevalence of PIU will be similar to that of other studies in the Spanish context [4,5,24,25]; b) there will be significant and inverse relationships between PIU and HRQoL; c) the greater the level of PIU, especially if it is severe, the worse will be HRQoL [54]; d) when inspecting the profiles, at least three different patterns will emerge (no problem, moderate problem, and severe problem) [5,28,30].

These four hypotheses are closely related to all analyses (except for the GPIUS-2 Cut-off point, line 336 et seq.). The authors have introduced a fifth hypothesis in relation to this point. Initially, we do not consider it because there is hardly any other research on this.

The study had the following hypotheses: a) the prevalence of PIU will be similar to that of other studies in the Spanish context [4,5,24,25]; b) there will be significant and inverse relationships between PIU and HRQoL; c) the greater the level of PIU, especially if it is severe, the worse will be HRQoL [54]; d) when inspecting the profiles, at least three different patterns will emerge (no problem, moderate problem, and severe problem) [5,28,30]; e) a diagnostic cut-off point for problematic Internet users will be proposed for the GPIUS2, in the same vein as other questionnaires [30].

In the current version, all hypotheses are discussed.

Reviewer #1: I am worry about the statement that the “ sample is representative”  here. The sample may be very representive to the study site (or region). However, I believe the finding of this study is not only apply to this region. The real population applying the finding, is beyond the region. A bar-chart is recommended for the post-hoc analysis. The result of pairwise comparison could be a difficult to have a clear and concise impression.

Authors The authors have explained the sampling process in more detail (line 167—170).

The sample obtained in the study is representative of students enrolled in public and concerted centers. With a 99% confidence level, a sampling error of 1% for a population variance of 0.50, the representative sample comprises 11,853 participants. In addition, sample selection was random, taking into account the number of students of each stage, the proportionality of centers in each location, and the type of center

The following sentence has been incorporated into the limitations (line 417—418). Thank you for your comment.

Although the sample is broad and representative of a Spanish region, we must be cautious when extrapolating the results to other cultural contexts.

Reviewer 2 Report

Thanks for giving me the chance to review the manuscript“Profiles of problematic Internet use and its impact on adolescents' health-related quality of life”. I think it is interesting and well-written. The topic is timely, and method/result sections are sound. Nevertheless, I have some further minor amendments.

It should be stressed that in the literature also other terms than problematic Internet use can be found (such as Internet addiction, Internet use disorder, etc.) and that all of these terms need to be compared and defined exactly.

The review of the literature in this manuscript is not full focus on the relationship between problematic Internet use and adolescents' health-related quality of life. It does not make a comprehensive and in-depth analysis of the literature in these two issues. It does not put forward the possible internal mechanism of the problematic Internet use and adolescents' health-related quality of life based on literature review. So, the inherent logic of study is not rigorous.

In Instruments section,the psychometric indicators of the two questionnaires in this study need to be reported,such as Cronbach's alpha.

The discussion section also needs to add some further information. It is suggested to add the discussion of the relationship between problematic Internet use and adolescents' health-related quality of life, and need to discuss the possible causes of why adolescents occur this phenomenon and further the potential mechanism of the between problematic Internet use and adolescents' health-related quality of life.

Author Response

REVIEWER COMMENTS FOR THE AUTHOR

Reviewer #2:

Reviewer #2 →  Thanks for giving me the chance to review the manuscript “Profiles of problematic Internet use and its impact on adolescents' health-related quality of life”. I think it is interesting and well-written. The topic is timely, and method/result sections are sound. Nevertheless, I have some further minor amendments.

Authors Thank you for your general assessment of the manuscript.

Reviewer #2 →  It should be stressed that in the literature also other terms than problematic Internet use can be found (such as Internet addiction, Internet use disorder, etc.) and that all of these terms need to be compared and defined exactly.

Authors Thank you for your comment. The authors initially considered this approach, but it was ruled out in the final version, so as not to introduce too many elements that could distract the reader from the object of the manuscript. However, we believe that this idea is generally reflected in the following point:

Currently, the DSM-5 [7] does not include Internet addiction or intensive Internet use as an addictive disorder, nor does it include Internet consumption in general within the "behavioral addictions.”  Notwithstanding, there have been several important approaches to Internet use in the last two decades. Thus, Tokunaga and Rains [8] refer to three great traditions such as: a) problematic Internet use (PIU) as an impulse control disorder (these authors have conceptualized it as a product of inadequate control over impulsive thoughts or feelings); b) PIU as a substance dependence (this has been regarded as analogous to a chemical addiction and is defined as Internet dependency); and c) PIU as a deficit of relational and relationship-building resources (negative effects of this Internet use in the person's life).

Also, Tokunaga and Rains (2016) made a thorough review to determine the different approaches, if any reader is interested. Only the three usual approaches under the term of Problematic Internet Use (PIU) are of interest to us. In most of the document, we preferred to use studies from one of these perspectives, to be consistent with the presentation of the theoretical antecedents.

Reviewer #2 →  The review of the literature in this manuscript is not full focus on the relationship between problematic Internet use and adolescents' health-related quality of life. It does not make a comprehensive and in-depth analysis of the literature in these two issues. It does not put forward the possible internal mechanism of the problematic Internet use and adolescents' health-related quality of life based on literature review. So, the inherent logic of study is not rigorous.

Authors Thank you for your comment. It is true that the introduction is very extensive and tries to explain all the necessary aspects, but the relationship between PIU and HRQoL could have been improved. In the new version, the following text has been added with several new references (see lines 136-151).

Therefore, at present, PIU has hardly been related to HRQoL. Although there is evidence that adaptive Internet use has positive effects on children's psychological well-being [48,49], currently research focuses on problems related to problematic, addictive, or maladaptive use related to the loss of HRQoL or of psychological well-being. Despite the evidence that has been found, there is currently great heterogeneity in the way we define and evaluate the two constructs. Thus, lower HRQoL has been reported in adolescents and young adults with PIU than in those without PIU [50,51]. This relationship between inadequate Internet use and poorer HRQoL has also been observed in primary education students [52]. There is evidence of the inverse relationship between PIU and HRQoL [53] and between Internet addiction and HRQoL [54]. Other studies have explored the use of social media, and online games are related to poorer psychological well-being, especially among girls [55]. From Caplan's approach to PIU, it could be considered that online social preference and mood regulation through the Internet generate social interactions that provoke feelings of emotional loneliness even if the individual apparently has social support, because this kind of communication lacks the sensory richness that human beings need [56,57]. This is also supported by studies indicating the two-way relationship between emotional loneliness and PIU, especially when the use of social media is predominant [58,59].

Reviewer #2 →  In Instruments section the psychometric indicators of the two questionnaires in this study need to be reported,such as Cronbach's alpha.

Authors → The reliability indicators (Cronbach alphas and omega coefficients) are indicated in Table 5 in the original version. Nothing has been changed from this point. You can see them on the right side of the table. It is indicated in the manuscript in lines 177 and 189.

Reviewer #2 →  The discussion section also needs to add some further information. It is suggested to add the discussion of the relationship between problematic Internet use and adolescents' health-related quality of life, and need to discuss the possible causes of why adolescents occur this phenomenon and further the potential mechanism of the between problematic Internet use and adolescents' health-related quality of life.

Authors → Changes have been incorporated into the discussion. See lines 365-367, 383-385, 421-423.

Reviewer 3 Report

This is an interesting and relevant manuscript. I have the following comments.

In the introduction, note that that generalized problematic internet use reflects mostly social networking (Costa et al., 2018; Problematic internet use and feelings of loneliness; Montag et al., 2015; Is it meaningful to distinguish between generalized and specific internet addiction? Evidence from a cross-cultural study from Germany, Sweden, Taiwan and China).

In the introduction and in the discussion, when referring to problems that online communication entails, it should be noted that some research has been raising concerns that online social interactions cause feelings of emotional loneliness even in the presence of apparently adequate social support, which might be due to online communication being devoid of the sensory richness that brains need to generate feelings of relatedness. As such it may cause loneliness even in those who have no objective reasons to feel lonely. See this issue discussed by Costa et al. (2018; Problematic internet use and feelings of loneliness), and supported by studies indicating a bidirectional causal nexus between emotional loneliness and problematic internet use (reflecting mostly social networking, as noted above) independently of social support (Yao and Zhong; 2014; Loneliness, social contacts and internet addiction; Zhang et al. (2018;. Relationships between social support, loneliness, and internet addiction postsecondary students: a longitudinal cross-lagged analysis. A bidirectional causal nexus between emotional loneliness and problematic internet use was also found by Tian et al. (2017; Associations between psychosocial factors and generalized pathological internet use in Chinese university students: A longitudinal cross-lagged analysis.

How sampling was randomized?

When discussing that girls were found to be have more problems than boys, note that the GPIUS measures generalized problematic internet use, which reflects mostly social networking (Montag et al., 2015; Is it meaningful to distinguish between generalized and specific internet addiction? Evidence from a cross-cultural study from Germany, Sweden, Taiwan and China).

The use of boys, although not exempt of social networking, likely encompasses more gamming.

Mood regulators are likely more than entertainment users, as too much mood regulation through external activities is linked to psychopatholoy.

Author Response

REVIEWER COMMENTS FOR THE AUTHOR

Reviewer #3:

Reviewer #3 This is an interesting and relevant manuscript. I have the following comments.

Authors The authors would like to thank you very much for your constructive comments. All references that you mention were very relevant and we have included them in the new version of the manuscript. Thank you for your input, as we think that it has enhanced the value of the manuscript.

Reviewer #3 In the introduction and in the discussion, when referring to problems that online communication entails, it should be noted that some research has been raising concerns that online social interactions cause feelings of emotional loneliness even in the presence of apparently adequate social support, which might be due to online communication being devoid of the sensory richness that brains need to generate feelings of relatedness. As such it may cause loneliness even in those who have no objective reasons to feel lonely. See this issue discussed by Costa et al. (2018; Problematic internet use and feelings of loneliness), and supported by studies indicating a bidirectional causal nexus between emotional loneliness and problematic internet use (reflecting mostly social networking, as noted above) independently of social support (Yao and Zhong; 2014; Loneliness, social contacts and internet addiction; Zhang et al. (2018;. Relationships between social support, loneliness, and internet addiction postsecondary students: a longitudinal cross-lagged analysis. A bidirectional causal nexus between emotional loneliness and problematic internet use was also found by Tian et al. (2017; Associations between psychosocial factors and generalized pathological internet use in Chinese university students: A longitudinal cross-lagged analysis.

Authors Thank you very much for your explanation. We have made several changes to the introduction including your suggestions (see lines 137-151).

Therefore, at present, PIU has hardly been related to HRQoL. Although there is evidence that adaptive Internet use has positive effects on children's psychological well-being [48,49], currently research focuses on problems related to problematic, addictive, or maladaptive use related to the loss of HRQoL or of psychological well-being. Despite the evidence that has been found, there is currently great heterogeneity in the way we define and evaluate the two constructs. Thus, lower HRQoL has been reported in adolescents and young adults with PIU than in those without PIU [50,51]. This relationship between inadequate Internet use and poorer HRQoL has also been observed in primary education students [52]. There is evidence of the inverse relationship between PIU and HRQoL [53] and between Internet addiction and HRQoL [54]. Other studies have explored the use of social media, and online games are related to poorer psychological well-being, especially among girls [55]. From Caplan's approach to PIU, it could be considered that online social preference and mood regulation through the Internet generate social interactions that provoke feelings of emotional loneliness even if the individual apparently has social support, because this kind of communication lacks the sensory richness that human beings need [56,57]. This is also supported by studies indicating the two-way relationship between emotional loneliness and PIU, especially when the use of social media is predominant [58,59].

Reviewer #3 ? How was sampling randomized?

Authors The authors have explained the sampling process in more detail (line 167—170).

The sample obtained in the study is representative of students enrolled in public and concerted centers. With a 99% confidence level, a sampling error of 1% for a population variance of 0.50, the representative sample comprises 11,853 participants. In addition, sample selection was random, taking into account the number of students of each stage, the proportionality of centers in each location, and the type of center

Reviewer #3 When discussing that girls were found to be have more problems than boys, note that the GPIUS measures generalized problematic internet use, which reflects mostly social networking (Montag et al., 2015; Is it meaningful to distinguish between generalized and specific internet addiction? Evidence from a cross-cultural study from Germany, Sweden, Taiwan and China).

The use of boys, although not exempt of social networking, likely encompasses more gamming.

Mood regulators are likely more than entertainment users, as too much mood regulation through external activities is linked to psychopathology.

Authors Thank you for your comment.

Changes have been incorporated into the discussion. See lines 365-367, 383-385, 421-423.

Reviewer 4 Report

introduction: 41-42 explain better

64: concept of the mood regulation is subjective and difficult to auto-assess in so young pepole

76: reference? 

125-130: explain better the link,   

Overall may be usefull to re-editing some phrases

Please check the references editing 

Author Response

REVIEWER COMMENTS FOR THE AUTHOR

Reviewer 4:

Reviewer #4 - introduction: 41-42 explain better

Authors Thank you for your comment.  This point has been explained in more detail.

Currently, adolescents live in a very different way from other generations, and young people have a complex and bidirectional relationship between what is occurring online and offline, and, according to the model of co-construction, adolescents shape their reality by connecting their offline and online worlds, the latter frequently predominating  [1].

Reviewer #4 64: concept of the mood regulation is subjective and difficult to auto-assess in so young pepole.

Authors Your comment is reasonable. Throughout the introduction, we explain the tools used for PIU (for us, especially the GPIUS-2), which has been widely used with adolescent population.

Machimbarrena, J.M.; Calgo, E.; Fernández-González, L.; Alvarez-Bardón, A.; Alvarez-Fernández, L.; Gonzalez-Cabrera, J. Internet Risks: An Overview of Victimization in Cyberbullying, Cyber Dating Abuse, Sexting, Online Grooming and Problematic Internet Use. Int. J. Environ. Beef. Public Health 2018.

Gámez-Guadix, M.; Orue, I.; Calgo, E. Evaluation of the Cognitive-behavioral Model of Generalized and Problematic Internet Use in Spanish adolescents. Psychothema 2013, 25, 299–306.

Yudes-Gómez, C.; Baridon-Chauvie, D.; Gonzalez-Cabrera, J.-M. Cyberbullying and Problematic Internet Use in Colombia, Uruguay and Spain: Cross-cultural Study. Comunicar 2018, 26, 2018–7.

In addition, the KIDSCREEEN is an evaluation tool for HRQoL, especially created for the stage between 8-18 years.

The use of self-reports has also been included in the limitations.

Reviewer #4 76: reference?

Authors Thanks for the comment. These two lines have been removed

Reviewer #4 125-130: explain better the link,  ,  

AuthorsThank you for your comment. This point has been explained in more detail.

Health-related quality of life (HRQoL) is a complex construct for which there is no accepted definition [40]. Nevertheless, there is a clear consensus to not define it as an absence of disease or disorder but instead from a more holistic point of view that integrates physical, psychological, emotional, and social aspects. Moreover, well-being must be perceived by the subject him- or herself and also by those around him or her [40–42].

Reviewer #4 - Overall may be usefull to re-editing some phrases

Authors The manuscript has been extensively reviewed by a native English speaker who holds a PhD in Psychology.

Reviewer #4 - Please check the references editing

Authors Thank you for your comment. We have checked the references in this new version 

Round 2

Reviewer 1 Report

Re: ijerph-597427, Profiles of problematic internet use and its impact on adolescents' health-related quality of life

        The authors have responded to my comments. I agree with the revision part of the representative sampling. However, I am still concerned about two points as follows:

My first comment is the need of employing the LPA here. As the authors stated in the response letter, that LPA may help to bring new insight from empirical data. I think this information should be addressed in the text, for readers to understand the reason why employing this method here. About the hypotheses here, there might be some difficulties to conduct the formal statistical testing (calculating the p-value to assert the rate of type I error). For example the hypothesis A cannot be tested, the cut-off point also cannot be tested with statistical hypothesis testing. I suggest the authors revise the hypotheses, refocus on hypotheses testing, and report the exact p-value for each hypothesis.

Author Response

Reviewer #1: My first comment is the need of employing the LPA here. As the authors stated in the response letter, that LPA may help to bring new insight from empirical data. I think this information should be addressed in the text, for readers to understand the reason why employing this method here.

Authors Thank you for your general appraisal of the review. We have incorporated information for readers about the LPA both in the method and the discussion.

2.4. Data analysis
…..3) Latent Profile Analysis (LPA) to identify profiles of Internet use, taking into account the five dimensions of the GPIUS2; for this purpose, exploratorily, we compared different solutions according to the Bayesian Criterion Information (BIC) and the Akaike Information Criterion (AIC). Latent Profile Analysis (LPA) tries to identify clusters of individuals based on responses to a series of continuous variables. LPA assumes that there are unobserved latent profiles that generate patterns of responses on indicator items. The resulting profiles can then inform behavioral or other important outcome differences and help explain inconsistencies in prior research [66];
4. Discussion
….At the same time, as hypothesized, LPA identified four subgroups, three of which were expected: non-problematic use, problematic use, and severe problematic use. However, a profile emerged that was called "mood regulator" which presents high scores in the dimension of mood regulation and online preference and low scores in the remaining dimensions. This is evidence of the interest in applying new statistical techniques to already established knowledge [66].

Reviewer #1: About the hypotheses here, there might be some difficulties to conduct the formal statistical testing (calculating the p-value to assert the rate of type I error). For example the hypothesis A cannot be tested, the cut-off point also cannot be tested with statistical hypothesis testing. I suggest the authors revise the hypotheses, refocus on hypotheses testing, and report the exact p-value for each hypothesis.

Authors We understand your concern. Certainly, from a rigorous point of view, hypotheses should be proposed so they can be falsified. However, in the fields of Social Sciences (psychology), hypotheses can also be provided as a provisional proposal that is not intended to be strictly demonstrated. In this case, the level of veracity granted will depend on the extent to which the empirical data support what is stated (e.g., this is common when comparing prevalences from different studies). This approach is applicable to Hypotheses "a" and "d" (Hypothesis "e" has been removed, as we think that, in the previous review, we did not understand your comment correctly).  On the other hand, it is possible to conduct formal statistical testing (calculating the p-value to assert the rate of Type I error) in hypotheses b and c.

In the rest of the hypotheses, we wanted to relate the current empirical data and contribute to current scientific knowledge with our evidence.  We have revised the wording to be less conclusive and more accurate. In this regard, we have also included an additional limitation about hypotheses a and d, along with the need for further studies in the future.

Reviewer 2 Report

The revised version had better answered  the issues I mentioned, and the quality of the manuscript has improved significantly.

Author Response

Thank you for your general appraisal of the review.

Thank you for your time.